# Diffusion Language Model Parallel Decoding via Product-of-Experts Bridge

**Juntong Shi** [1]  **Brian L. Trippe** [1]  **Jure Leskovec** [1]  **Stefano Ermon** [1]  **Minkai Xu** [1]

## Abstract

Diffusion language models (DLMs) offer substantial speed advantages through parallel decoding, but the lack of token dependencies limits generation quality compared to autoregressive (AR) models. Recent progress attempts to bridge the gap via importance sampling, with DLM being the proposal and AR being the target. However, due to the huge gap between their distributions, the sampling requires a large number of particles and is thus expensive to compute. In this paper, we introduce PoE-Bridge, a novel decoding framework that drastically improves generation speed and accuracy by introducing an intermediate distribution to bridge the gap. The distribution is constructed as a Product-of-Experts (PoE) of the DLM proposal and the AR target. With the intermediate distribution, we first use the DLM to draft multiple continuations in parallel, then apply rejection sampling to verify the drafted tokens and move the resulting candidates toward the PoE. We then use importance sampling to further correct the PoE-aligned candidates toward the AR target. We further propose several improved techniques, including mixed-temperature sampling for enhanced diversity and elastic rejection windows for reducing wasted verification. Empirically, PoE-Bridge achieves significantly improved accuracy with $5\times$ speedup over the standard DLM decoding approach, and recovers at least 95% of the target AR model's performance, efficiently advancing most of the quality gap on challenging mathematical reasoning and coding tasks. Our code is available at https://github.com/juntongshi48/poe-bridge.

BT, JL, and SE co-supervised the project, secured resources, and contributed to the conceptualization and critical revision of the manuscript. [1]Stanford University. Correspondence to: Juntong Shi <juntong@stanford.edu>, Minkai Xu <minkai@cs.stanford.edu>.

*Proceedings of the 43$^{rd}$ International Conference on Machine Learning*, Seoul, South Korea. PMLR 306, 2026. Copyright 2026 by the author(s).

## 1. Introduction

Autoregressive (AR) language models remain the dominant approach to text generation (Vaswani et al., 2017; Wolf et al., 2020; Touvron et al., 2023), achieving strong performance on challenging tasks such as mathematical reasoning (Wei et al., 2022; Trinh et al., 2024; Yang et al., 2024) and code synthesis (Roziere et al., 2023; Hui et al., 2024). However, AR decoding is inherently sequential—generating tokens one by one in a strict left-to-right order. This sequential dependency inherently limits inference-time parallelism and makes latency and throughput a major bottleneck.

Diffusion language models (DLMs) (Sahoo et al., 2024; Shi et al., 2024; Ye et al., 2025; Nie et al., 2025), derived from the broader discrete diffusion framework (Austin et al., 2021b; Campbell et al., 2022; Lou et al., 2024), provide an appealing alternative by enabling parallel decoding. By generating and refining multiple tokens simultaneously, DLM can, in principle, produce entire sequences in a fraction of the sequential steps required by AR models. In practice, however, DLMs still lag behind strong AR models in generation quality. This gap largely stems from the conditional independence assumptions used to enable parallel decoding, under which tokens generated within the same step are modeled independently rather than jointly. As a result, the anticipated speedups from parallel generation are difficult to realize without incurring substantial degradation in output quality.

This gap suggests that effective parallel decoding requires a mechanism to inject inter-token dependencies into DLM drafts while preserving their parallel sampling efficiency. Notably, while sampling is expensive, AR models are comparatively efficient at *scoring* candidate continuations given a prefix. This asymmetry points out a natural direction to combine DLM's parallel decoding capacity with an external AR that evaluates candidate continuations and guides decoding toward faithful, high-quality generations. Recent efforts in this direction have resorted to Monte Carlo methods, such as rejection sampling (RS) and importance sampling (IS) (Israel et al., 2025; Xu et al., 2025). In these setups, a DLM acts as a fast proposal distribution whose outputs are subsequently verified or corrected by a stronger AR target model. However, in practical language generation tasks, both naive approaches break down due to the same fundamental reason:

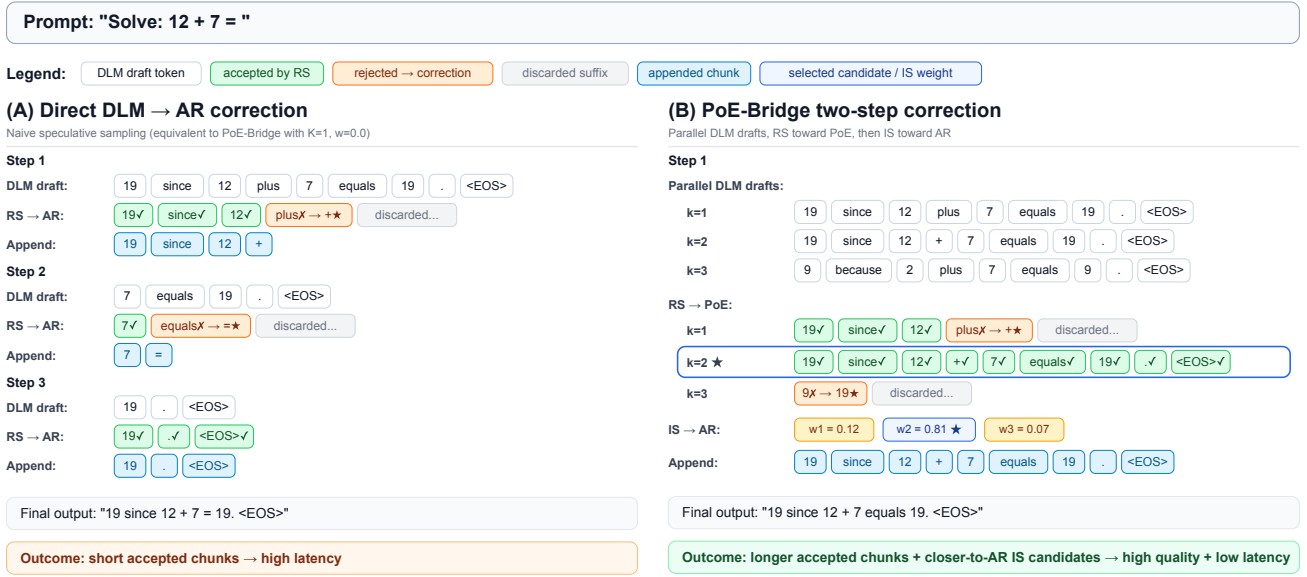

*Figure 1.* Comparison between naive speculative sampling and PoE-Bridge. **(A)** Naive speculative sampling directly corrects DLM drafts from $p_D$ to the AR target $p_{AR}$. Due to the large proposal–target mismatch, direct verification often accepts only short prefixes, resulting in limited throughput gains. **(B)** PoE-Bridge splits this difficult correction into two easier stages: speculative rejection sampling first moves DLM drafts toward the intermediate bridge distribution $p_{PoE}$, and importance sampling then refines the accepted candidates toward $p_{AR}$. Because $p_{PoE}$ is closer to both the DLM proposal and the AR target, PoE-Bridge achieves longer accepted chunks and more stable importance weights, enabling high generation quality and low latency.

the distribution mismatch between a parallel DLM proposal and a strong AR target is large. In rejection sampling, this mismatch leads to frequent rejection, causing the decoding process to collapse into near-sequential generation. While in importance sampling, it requires an impractically large number of candidates to obtain a good candidate; with a small candidate budget, resampling often selects merely the *least implausible* DLM continuation rather than a truly faithful AR sample.

In this work, we introduce PoE-Bridge, a novel decoding framework that can produce generations as faithful as AR decoding while preserving the parallel decoding advantages of DLMs. Our key innovation is to reduce the mismatch faced by Monte Carlo correction by introducing an intermediate *bridge* distribution between the DLM proposal and the AR target. Figure 1 illustrates this idea: instead of directly correcting DLM drafts toward the AR target, PoE-Bridge decomposes the difficult $p_D \rightarrow p_{AR}$ correction into two easier transitions through a PoE bridge distribution. Intuitively, this bridge is a Product-of-Experts (PoE) interpolation of the form (the precise, tractable definition over text sequences is given in Section 4.2):

$$p_{PoE}(\mathbf{x}) = \frac{1}{Z_w} p_D(\mathbf{x})^w p_{AR}(\mathbf{x})^{1-w},$$

where $w \in [0, 1]$ controls the position of the bridge. When $w$ is close to 1, the bridge stays close to the DLM proposal; when $w$ is close to 0, it approaches the AR target. In the method section, we instantiate this idea with a tractable

token-level PoE definition that supports efficient speculative verification and importance weighting. The bridge splits the difficulty DLM-to-AR correction into two easier steps: $p_D \rightarrow p_{PoE} \rightarrow p_{AR}$.

Leveraging this bridge, PoE-Bridge first filters candidate continuations generated by the DLM in parallel toward the PoE distribution using speculative rejection sampling. Since the bridge is deliberately constructed to be closer to the DLM proposal, it enables a large fraction of DLM-generated tokens to be accepted and thus preserving high parallel decoding throughput. Next, PoE-Bridge applies importance resampling to refine the filtered candidate continuations toward the AR target. Since the first stage already moves candidates closer to the AR distribution, the final importance-resampling step serves as a final lightweight refinement step rather than a primary mechanism for correcting the proposal–target mismatch, making it effective under realistic candidate budgets.

To fully realize the potential of PoE-Bridge, we incorporate several novel mechanisms tailored for practical LLM inference. First, to address the diversity-quality tradeoff in importance sampling, we introduce *Mixed-Temperature Sampling*. By stratifying proposal entropy across candidates, this strategy actively prevents mode collapse and ensures a diverse coverage of the target distribution, significantly improving the effective sample size. Second, to reduce wasted verification compute caused by fixed-length drafting windows, we propose an *Elastic Rejection Window*,

which adaptively adjusts the proposal and verification scope based on the observed acceptance lengths. This mechanism reduces verification over tokens that are unlikely to be accepted, which is especially helpful in our high-throughput decoding setting where multiple candidates are processed in parallel.

Empirically, PoE-Bridge demonstrates a "best-of-both-worlds" performance. On challenging mathematical reasoning and coding benchmarks, PoE-Bridge achieves substantially higher accuracy while delivering up to a $5\times$ speedup over standard DLM decoding. Meanwhile, it recovers at least $95\%$ of the target AR model's accuracy while achieving $2\times$ the throughput of AR decoding. Our results demonstrate that AR-level faithfulness can be approached without giving up the parallel decoding benefits of diffusion models.

## 2. Related Works

**Rejection-based decoding acceleration.** Speculative decoding (Leviathan et al., 2023; Chen et al., 2023; Narasimhan et al., 2025) uses a fast draft model to propose multiple tokens that are verified by a stronger AR model, yielding samples faithful to the target distribution. Extensions consider alternative draft models, including any-order AR and diffusion LMs (Guo & Ermon, 2025; Christopher et al., 2025; Xiao et al., 2024), but do not address the regime where a *large* DLM is the primary parallel proposal, with substantial mismatch and verification overhead. Adaptive Parallel Decoding (APD) (Israel et al., 2025) adopts a rejection-style procedure for large DLMs with a lightweight verifier, but its design targets speed rather than faithful sampling from a strong AR expert, leaving performance bounded by the DLM. Liu et al. (2025); Kumar et al. (2026) primarily accelerate speculative sampling by overlapping drafting and verification, while PoE-Bridge improves efficiency by increasing acceptance through mixing probabilities. These directions are complementary and could be combined to further improve throughput.

**Importance sampling and reranking for correcting parallel generation.** A complementary line of work applies importance sampling (IS), reranking, or energy-based corrections to mitigate independence-induced errors in non-autoregressive or diffusion-style generation (Xu et al., 2025; Gu et al., 2018; Ghazvininejad et al., 2019; Gu & Kong, 2021; Savinov et al., 2022; Zheng et al., 2023). These methods use IS-style corrections to encourage token dependencies during parallel decoding, but generally provide only approximate alignment to the target distribution and are sensitive to proposal–target mismatch and limited compute. Moreover, existing studies on diffusion language models (Xu et al., 2025) focus on relatively small models and do not examine settings where a large DLM performs most of the generation in parallel.

## 3. Preliminaries

### 3.1. Discrete Diffusion Language Models

Discrete diffusion models (DLMs) (Austin et al., 2021a; Campbell et al., 2022; Lou et al., 2024) offer an alternative to the autoregressive generation by producing text through iterative refinement of a partially corrupted sequence, rather than committing tokens one by one sequentially. Among them, masked diffusion models show the most impressive results. They define a forward noising process that gradually replaces tokens with the special mask token $\mathbf{m}$, and learn a reverse model that recovers the masked content.

**Masked Discrete Diffusion.** Let $\mathbf{x} = (\mathbf{x}_1, \ldots, \mathbf{x}_n)$ be a sequence of $n$ tokens and $\mathbf{x}^0$ the clean sequence. The forward diffusion process $q$ independently masks each token with probability $1 - \alpha(t)$ at time $t$, where $\alpha : [0, 1] \to [0, 1]$ is a monotonically decreasing noise schedule with $\alpha(0) = 1$ and $\alpha(1) = 0$. The resulting distribution factorizes as $q(\mathbf{x}^t \mid \mathbf{x}^0) = \prod_{i=1}^n q(\mathbf{x}_i^t \mid \mathbf{x}_i^0)$, with

$$q(\mathbf{x}_i^t \mid \mathbf{x}_i^0) = \begin{cases} 1 - \alpha(t), & \text{if } \mathbf{x}_i^t = \mathbf{m} \\ \alpha(t), & \text{if } \mathbf{x}_i^t = \mathbf{x}_i^0. \end{cases} \quad (1)$$

The denoising network $\boldsymbol{\mu}_\theta(\mathbf{x} \mid \mathbf{x}^t; \theta)$ is trained to predict the masked tokens from the visible context. It adopts a mean-field parameterization, predicting each position independently: $\boldsymbol{\mu}_\theta(\mathbf{x} \mid \mathbf{x}^t; \theta) = \prod_i \boldsymbol{\mu}_\theta(\mathbf{x}_i \mid \mathbf{x}^t; \theta)$. Let $p_D$ denote the induced generation distribution by running the DLM decoder with $\boldsymbol{\mu}_\theta$. Training maximizes a variational lower bound (Shi et al., 2024; Sahoo et al., 2024):

$$\begin{aligned} \log p_D(\mathbf{x}^0) \geq \mathbb{E}_{t \sim U(0,1), \mathbf{x}^t \sim q(\mathbf{x}^t \mid \mathbf{x}^0)} w(t) \\ \left[ \log \boldsymbol{\mu}_\theta(\mathbf{x}_{\{i: \mathbf{x}_i^t = \mathbf{m}\}}^0 \mid \mathbf{x}_{\{i: \mathbf{x}_i^t \neq \mathbf{m}\}}^t; \theta) \right], \end{aligned} \quad (2)$$

where $w(t) = \frac{-\alpha'(t)}{1 - \alpha(t)}$ is a known reweighting term determined by the noise schedule.

To generate, DLMs start from a fully masked sequence and progressively unmask tokens. Progressing from time $t$ to $s < t$, already-unmasked tokens are kept fixed, while each masked token is either kept masked or revealed using the model prediction. Formally, for each position $i$,

$$p_D(\mathbf{x}_i^s \mid \mathbf{x}^t) = \begin{cases} \frac{1 - \alpha_s}{1 - \alpha_t}, & \text{if } \mathbf{x}_i^t = \mathbf{m}, \ \mathbf{x}_i^s = \mathbf{m}, \\ \frac{\alpha_s - \alpha_t}{1 - \alpha_t} \boldsymbol{\mu}_\theta(\mathbf{x}_i^s \mid \mathbf{x}^t; \theta), & \text{if } \mathbf{x}_i^t = \mathbf{m}, \ \mathbf{x}_i^s \neq \mathbf{m}, \\ \delta(\mathbf{x}_i^s = \mathbf{x}_i^t), & \text{if } \mathbf{x}_i^t \neq \mathbf{m}. \end{cases} \quad (3)$$

**Decoding Order.** The generic DLM decoder above allows arbitrary subsets of masked positions to be revealed at each step. In this work, we instead use a semi-autoregressive left-to-right decoding order at inference time, so that each DLM draft forms a contiguous continuation from the current prefix. This choice makes the DLM draft probabilities token-wise comparable with standard autoregressive left-to-right

likelihoods, which will be useful for the decoding procedure introduced later. It is also consistent with prior observations that practical DLM decoding heuristics (Ye et al., 2025; Nie et al., 2025) often induce approximately left-to-right generation patterns in practice (Gong et al., 2025), and that explicitly enforcing such an order can improve generation quality (Israel et al., 2025).

Conceptually, we define $p_D$ as the autoregressive distribution induced by decoding the DLM one token at a time:

$$p_D(\mathbf{x}_i \mid \mathbf{x}_{<i}) := \boldsymbol{\mu}_\theta(\mathbf{x}_i \mid \mathbf{x}_{<i}),$$
$$p_D(\mathbf{x}) := \prod_{i=1}^{n} p_D(\mathbf{x}_i \mid \mathbf{x}_{<i}). \tag{4}$$

This definition places the DLM distribution in the same left-to-right factorized form as a standard autoregressive language model. For comparison, we denote the distribution of the target autoregressive model by $p_{AR}(\mathbf{x}) = \prod_{i=1}^{n} p_{AR}(\mathbf{x}_i \mid \mathbf{x}_{<i})$.

Importantly, $p_D$ is only a conceptual definition. Sampling from and evaluating the exact conditional $p_D(\mathbf{x}_i \mid \mathbf{x}_{<i})$ requires rerunning the DLM after every newly generated token. To obtain an efficient decoding procedure, we instead approximate the sequential autoregressive DLM distribution using DLM's mean-field chunk parameterization. Let $c_t$ denote the index of the first token decoded at step $t$, with $c_1 = 1$ and $c_{T+1} = n + 1$. At decoding step $t$, we run the DLM once conditioned on the current prefix $\mathbf{x}_{<c_t}$, and predict the entire chunk $\mathbf{x}_{c_t:c_{t+1}-1}$ in parallel. For each token $i \in [c_t, c_{t+1})$, we approximate the exact autoregressive conditional with

$$p_D(\mathbf{x}_i \mid \mathbf{x}_{<i}) \approx \tilde{p}_D(\mathbf{x}_i \mid \mathbf{x}_{<i}) := \boldsymbol{\mu}_\theta(\mathbf{x}_i \mid \mathbf{x}_{<c_t}). \tag{5}$$

That is, instead of conditioning on the progressively extended prefix $\mathbf{x}_{<i}$, all tokens within the chunk share the same conditioning prefix $\mathbf{x}_{<c_t}$.

Under this approximation, the chunk probability becomes:

$$p_D(\mathbf{x}_{c_t:c_{t+1}-1} \mid \mathbf{x}_{<c_t}) \approx \tilde{p}_D(\mathbf{x}_{c_t:c_{t+1}-1} \mid \mathbf{x}_{<c_t})$$
$$:= \prod_{i=c_t}^{c_{t+1}-1} \boldsymbol{\mu}_\theta(\mathbf{x}_i \mid \mathbf{x}_{<c_t}). \tag{6}$$

which can be sampled from and evaluated in parallel using a single DLM forward pass.

We use these tractable approximations throughout our decoding method.

### 3.2. Product of Experts

Product of Experts (PoE) (Hinton, 1999) combines multiple probabilistic "experts" by multiplying their densities

with different weights and renormalizing. Given experts $\{p_j(\mathbf{x})\}_{j=1}^{n}$ and non-negative weights $\{w_j\}_{j=1}^{n}$, the PoE distribution is

$$p_{\text{PoE}}(\mathbf{x}) = \frac{1}{Z} \prod_{j=1}^{m} p_j(\mathbf{x})^{w_j}, \qquad Z = \int \prod_{j=1}^{m} p_j(\mathbf{x})^{w_j} d\mathbf{x}. \tag{7}$$

PoE models are often easy to evaluate up to a normalizing constant but nontrivial to sample from. A common strategy is to first sample from a tractable proposal distribution and use Monte Carlo correction to move samples toward the target PoE distribution. Two common correction mechanisms are (i) rejection sampling or (ii) importance sampling.

**Rejection Sampling.** Let $q(\mathbf{x})$ be a tractable proposal distribution and $\pi(\mathbf{x})$ be the target distribution. Classical rejection sampling assumes the existence of a domination constant $M$ such that $\pi(\mathbf{x}) \leq Mq(\mathbf{x})$ for all $\mathbf{x}$. It draws a proposal sample $\mathbf{x} \sim q$ and accepts it with probability

$$a(\mathbf{x}) = \frac{\pi(\mathbf{x})}{Mq(\mathbf{x})}. \tag{8}$$

When such an $M$ is available, accepted samples are exactly distributed according to $\pi$. However, finding a tight and valid $M$ is often difficult in high-dimensional sequence spaces, and a loose $M$ leads to low acceptance.

A common rejection-style alternative is acceptance-complement sampling (Devroye, 1986), which avoids the need for a global domination constant. Instead, it accepts samples according to the direct proposal–target density ratio and compensates for the resulting bias by drawing a correction sample from a residual distribution upon rejection. Importantly, this residual distribution depends on the target density, which is typically more expensive to evaluate or sample from than the proposal.

**Importance Sampling.** Importance sampling offers an alternative correction mechanism that avoids rejection. Given candidate samples $\mathbf{x}^{(k)} \sim q$, importance sampling assigns each sample a weight proportional to $\pi(x^{(k)})/q(x^{(k)})$ and uses these weights to resample from the candidate set. This procedure ensures that candidates are selected in proportion to their likelihood under the target distribution $\pi$. Unlike rejection sampling, which guarantees exact samples from $\pi$, importance sampling is only asymptotically correct. With a finite candidate set, the resampling step introduces bias, whose magnitude depends on the proposal–target mismatch and the available sample budget.

## 4. Method

### 4.1. Overview

**Problem statement.** We consider the following problem setting. We are given two language model experts: a diffu-

---

**Algorithm 1** PoE-Bridge Decoding

1: **Input:** DLM denoiser $\boldsymbol{\mu}_\theta$, AR model $p_{\text{AR}}$, PoE weight $w$, #candidates $K$, proposal temperatures $\{\tau_k\}$, target temperature $\tau_\star$, elastic rejection window size $V$, max length $n$, prompt $\mathbf{p}$
2: $\mathbf{x} \leftarrow \mathbf{p}$
3: **while** $|\mathbf{x}| < n$ **and** EOS $\notin \mathbf{x}$ **do**
4:      $p^D_{|\mathbf{x}|+1:n} \leftarrow \boldsymbol{\mu}_\theta(\mathbf{x}^D_{|\mathbf{x}|+1:n} \mid \mathbf{x}_{\leq|\mathbf{x}|})$
5:      **while** $|\mathbf{x}| < n$ **and** EOS $\notin \mathbf{x}$ **do**
         ▷ {$c$: next position; $m$: true window size}
6:          $c, m \leftarrow |\mathbf{x}| + 1, \min(V, n - |\mathbf{x}|)$
7:          $r \leftarrow c + m - 1$
         ▷ {Draw $K$ candidates in parallel}
8:          **for** $k = 1, \ldots, K$ **in parallel do**
9:              $\hat{\mathbf{x}}^{(k)}_{c:r} \leftarrow \texttt{sample}(p^D_{c:r}, \tau_k)$
10:         $p^{\text{AR},(k)}_{c:r} \leftarrow p_{\text{AR}}(\hat{\mathbf{x}}^{(k)}_{c:r} \mid \mathbf{x}_{<c})$
11:         $p^{\text{PoE},(k)}_{c:r} \leftarrow$
            $\texttt{norm}\left((p^D_{c:r})^w \odot (p^{\text{AR},(k)}_{c:r})^{1-w}, \text{dim}{=}{-}1\right)$
12:         $(\hat{\mathbf{x}}^{(k)}_{c:c+a_k-1}, a_k) \leftarrow$
            $\texttt{SpecReject}\left(\hat{\mathbf{x}}^{(k)}_{c:r}, p^D_{c:r}, p^{\text{PoE},(k)}_{c:r}\right)$
            ▷ {SpecReject follows Eqs. 11 and 12}
         ▷ {Compute importance sampling weights}
13:         $p^{(k)}_{\text{AR}} \leftarrow \prod_{i=c}^{c+a_k-1} p^{\text{AR},(k)}_i[\hat{\mathbf{x}}^{(k)}_i]$
14:         $p^{(k)}_{\text{PoE}} \leftarrow \prod_{i=c}^{c+a_k-1} p^{\text{PoE},(k)}_i[\hat{\mathbf{x}}^{(k)}_i]$
15:         $w_k \leftarrow \left(p^{(k)}_{\text{AR}}\right)^{1/\tau_\star} / \left(p^{(k)}_{\text{PoE}}\right)^{1/\tau_k}$
16:         **end for**
17:         $k^\star \sim \text{Cat}(\texttt{norm}([w_1, \ldots, w_K]))$
18:         $\mathbf{x} \leftarrow \texttt{concat}(\mathbf{x}, \hat{\mathbf{x}}^{(k^\star)}_{c:c+a_{k^\star}-1})$
19:         **if** $a_{k^\star} < m$ **then break**
20:      **end while**
21: **end while**
22: **Output:** $\mathbf{x}$

---

sion language model (DLM) with generation distribution $\tilde{p}_D$ and an autoregressive (AR) model with $p_{\text{AR}}$. Given a prefix $\mathbf{x}_{<c}$, the DLM can generate a candidate continuation $\hat{\mathbf{x}}_{c:n}$ in parallel across multiple future positions. In contrast, while AR models can leverage KV caching to reduce the per-token FLOPs, it must still generate tokens sequentially, making long-horizon sampling latency bounded under high-parallelism on modern accelerators. However, the AR model can efficiently score a proposed continuation via $p_{\text{AR}}(\hat{\mathbf{x}}_{c:n} \mid \mathbf{x}_{<c})$, since teacher-forced likelihood evaluation parallelizes across positions. Our goal is to approximate faithful, high-quality samples from $p_{\text{AR}}(\mathbf{x}_{c:n} \mid \mathbf{x}_{<c})$ while relying on the DLM to perform most samplings.

This naturally suggests a Monte Carlo correction view: we use $\tilde{p}_D(\cdot \mid \mathbf{x}_{<c})$ as a tractable proposal distribution and set $p_{\text{AR}}(\cdot \mid \mathbf{x}_{<c})$ as the PoE target. Under this formulation,

two naive approaches follow from classical Monte Carlo methods—*rejection sampling* and *importance sampling*.

**Naive approaches.** A direct baseline is to apply speculative *rejection sampling*. Given a prefix $\mathbf{x}_{<c}$, we first sample a continuation $\hat{\mathbf{x}}_{c:n}$ in parallel from the DLM proposal $\tilde{p}_D(\cdot \mid \mathbf{x}_{<c})$, and then verify it left-to-right under the AR model, accepting tokens until the first rejection. Upon rejection, we restart from the rejection position and resample a new suffix from $\tilde{p}_D$. In practice, however, the huge discrepancy between $\tilde{p}_D$ and $p_{\text{AR}}$ leads to frequent rejection, so the accepted prefix is typically short. As shown later in Table 2, this direct DLM-to-AR correction setting ($w = 0.0$) suffers from frequent all-reject steps and short accepted prefixes, which substantially limit decoding speedup.

The alternative strategy is *importance sampling*. We sample $K$ candidate continuations $\{\hat{\mathbf{x}}^{(k)}_{c:n}\}_{k=1}^K \sim \tilde{p}_D(\cdot \mid \mathbf{x}_{<c})$ and resample one using unnormalized weights $W_k \propto \frac{p_{\text{AR}}(\hat{\mathbf{x}}^{(k)}_{c:n}|\mathbf{x}_{<c})}{\tilde{p}_D(\hat{\mathbf{x}}^{(k)}_{c:n}|\mathbf{x}_{<c})}$. This approach avoids sequential rejection and can be fully parallelized. However, when $\tilde{p}_D$ and $p_{\text{AR}}$ are poorly matched, samples from $\tilde{p}_D$ rarely fall in high-probability regions of $p_{\text{AR}}$. More intuitively, resampling tends to select the *least implausible* DLM continuation, rather than a truly high-quality AR continuation. Consequently, naive importance sampling provides no quality guarantee unless the candidate set is very large, which then becomes incompatible with the goal of efficient decoding.

### 4.2. PoE-Bridge Decoding

The inefficiency of rejection sampling and the lack of quality guarantees in naive importance sampling share a common root cause: the large distribution mismatch between the proposal $\tilde{p}_D$ and the target $p_{\text{AR}}$. This observation naturally motivates us to introduce an intermediate distribution that bridges the two. A natural bridge construction is the sequence-level PoE:

$$p_{\text{PoE}}(\mathbf{x}_{c:n}) \propto \tilde{p}_D(\mathbf{x}_{c:n})^w p_{\text{AR}}(\mathbf{x}_{c:n})^{1-w}, \quad (9)$$

with the PoE weight $w \in [0, 1]$ controlling the relative position of the PoE between the proposal and target.

However, directly sampling from or evaluating next-token probabilities under this distribution is generally intractable. Instead, we define the PoE bridge distribution autoregressively using a token-level PoE interpolation over next-token probabilities:

$$p_{\text{PoE}}(\mathbf{x}_i \mid \mathbf{x}_{<i}) := \frac{\tilde{p}_D(\mathbf{x}_i \mid \mathbf{x}_{<i})^w p_{\text{AR}}(\mathbf{x}_i \mid \mathbf{x}_{<i})^{1-w}}{Z_i(\mathbf{x}_{<i})},$$

$$p_{\text{PoE}}(\mathbf{x}_{c:n}) = \prod_{i=c}^{n} p_{\text{PoE}}(\mathbf{x}_i \mid \mathbf{x}_{<i}), \quad (10)$$

where $Z_i(\mathbf{x}_{<i}) = \sum_{\mathbf{x}_i} \tilde{p}_D(\mathbf{x}_i \mid \mathbf{x}_{<i})^w p_{\mathrm{AR}}(\mathbf{x}_i \mid \mathbf{x}_{<i})^{1-w}$ is the local normalizing constant at position $i$.

This token-level PoE definition brings two key benefits. First, because it uses the tractable DLM proposal $\tilde{p}_D$, it yields tractable next-token probabilities and chunk likelihoods that can be efficiently evaluated for speculative rejection and importance weighting. Second, like the sequence-level PoE in Equation (9), it still provides a smooth interpolation between the DLM proposal and the AR target. The interpolation weight $w$ controls the position of the bridge: $w = 1$ recovers $\tilde{p}_D$, while $w = 0$ recovers $p_{\mathrm{AR}}$. Intermediate values let $p_{\mathrm{PoE}}$ lie between the proposal and target, reducing the mismatch that undermines both naive rejection and importance sampling.

With this bridge distribution in place, we are able to effectively combine rejection and importance sampling, with both efficiency and quality guarantees.

**Speculative Rejection Sampling from $\tilde{p}_D$ to $p_{\mathrm{PoE}}$.** Given a prefix $\mathbf{x}_{<c}$, we first apply speculative rejection sampling, but instead of targeting $p_{\mathrm{AR}}$, we now target $p_{\mathrm{PoE}}$. Concretely, we first sample continuation tokens $\hat{\mathbf{x}}_{c:n}$ in parallel from $\tilde{p}_D$ and evaluate their likelihoods under both $\tilde{p}_D$ and $p_{\mathrm{AR}}$. Following Leviathan et al. (2023); Chen et al. (2023), we accept each proposed token with probability

$$\min\left(1, \frac{p_{\mathrm{PoE}}(\hat{\mathbf{x}}_i \mid \hat{\mathbf{x}}_{<i})}{\tilde{p}_D(\hat{\mathbf{x}}_i \mid \hat{\mathbf{x}}_{<i})}\right), \qquad (11)$$

and accept consecutive tokens until the first rejection. At the rejection position $c + a$, where $a$ is the number of consecutively accepted tokens, we resample a correction token from the residual distribution

$$\hat{\mathbf{x}}_{c+a} \sim \mathrm{norm}\left(\max\left(0,\ p_{\mathrm{PoE}}(\cdot \mid \hat{\mathbf{x}}_{<c+a}) - \tilde{p}_D(\cdot \mid \hat{\mathbf{x}}_{<c+a})\right)\right). \qquad (12)$$

This correction step compensates for the aggressive acceptance rule without a domination constant. As a result, each verified token is sampled from the bridge $p_{\mathrm{PoE}}$.

Since $p_{\mathrm{PoE}}$ interpolates between $\tilde{p}_D$ and $p_{\mathrm{AR}}$, the rejection step operates on a smaller proposal–target mismatch than direct DLM-to-AR correction. Thus, more draft tokens are accepted at each step, improving decoding throughput. The interpolation weight $w$ controls the tradeoff between acceptance rate and faithfulness to the AR target: larger $w$ places the bridge closer to the proposal, improving acceptance rates but requiring stronger importance resampling correction.

**Importance Sampling from $p_{\mathrm{PoE}}$ to $p_{\mathrm{AR}}$.** We repeat the speculative rejection sampling procedure in parallel to obtain $K$ candidate continuations $\{\hat{\mathbf{x}}_{c:c+a_k-1}^{(k)}\}_{k=1}^K$, where $a_k$ denotes the number of accepted tokens in candidate $k$ (including the correction token). Since these candidates are distributed under $p_{\mathrm{PoE}}$, we can apply importance sampling to further correct them toward the AR target $p_{\mathrm{AR}}$. Specifi-

cally, we assign each candidate a weight

$$w_k \propto \frac{p_{\mathrm{AR}}(\hat{\mathbf{x}}_{c:c+a_k-1}^{(k)} \mid \mathbf{x}_{<c})}{p_{\mathrm{PoE}}(\hat{\mathbf{x}}_{c:c+a_k-1}^{(k)} \mid \mathbf{x}_{<c})}. \qquad (13)$$

We then resample a single candidate according to the normalized weights.

This resampling procedure approximates sampling from $p_{\mathrm{AR}}$ as $K \to \infty$, though finite $K$ still introduces bias. However, compared to directly applying importance sampling from $\tilde{p}_D$, the bridge distribution substantially stabilizes the importance weights because $p_{\mathrm{PoE}}$ already moves samples closer to the AR target.

Finally, all stages of the procedure are highly parallelizable: candidate proposal and verification are parallelized across token positions; the $K$ candidates are generated independently in parallel; and importance-weight computation is inexpensive. Overall, the intermediate PoE bridge distributes the difficult DLM-to-AR correction across two easier stages, enabling both higher acceptance rates during speculative rejection and more stable importance weights during resampling.

In the following, we describe two practical techniques that make PoE-Bridge feasible in realistic inference settings.

### 4.3. Importance Sampling with Mixed-temperature

In modern LLM inference, low-temperature sampling is commonly used to stabilize generation. However, with a limited candidate budget $K$, drawing all candidates at a single low temperature can lead to mode collapse, producing nearly identical samples that reduce the effectiveness of importance sampling. To improve diversity under a fixed budget, we instead draw candidates from a family of tempered PoE distributions:

$$\mathbf{x}^{(k)} \sim q_{\tau_k}, \qquad q_{\tau_k}(\mathbf{x}) \propto p_{\mathrm{PoE}}(\mathbf{x})^{1/\tau_k}. \qquad (14)$$

The temperature schedule $\{\tau_k\}_{k=1}^K$ linearly spans $[\tau_{\mathrm{low}}, \tau_{\mathrm{high}}]$ (more details in Section A). Smaller $\tau_k$ concentrates samples in high-probability regions of $p_{\mathrm{PoE}}$, while larger $\tau_k$ promotes exploration.

Importantly, under our autoregressive token-level PoE definition in Equation (10), sampling from tempered PoE distributions remains tractable and easy to implement. Since $p_{\mathrm{PoE}}(\mathbf{x})$ factorizes across positions and each next-token PoE distribution is defined as a geometric interpolation between $\tilde{p}_D$ and $p_{\mathrm{AR}}$, temperature scaling preserves the same structure:

$$q_{\tau_k}(\mathbf{x}_i \mid \mathbf{x}_{<i}) \propto \tilde{p}_D(\mathbf{x}_i \mid \mathbf{x}_{<i})^{w/\tau_k} p_{\mathrm{AR}}(\mathbf{x}_i \mid \mathbf{x}_{<i})^{(1-w)/\tau_k},$$

$$q_{\tau_k}(\mathbf{x}_{c:n}) = \prod_{i=c}^{n} q_{\tau_k}(\mathbf{x}_i \mid \mathbf{x}_{<i}). \qquad (15)$$

*Table 1.* Benchmark results on GSM8K, MATH, HumanEval, and MBPP. Under each dataset, we report Accuracy (Acc.) and Throughput (Thrpt.) in tokens/second.

| Method | GSM8K | | MATH | | HumanEval | | MBPP | |
|---|---|---|---|---|---|---|---|---|
| | Acc.↑ | Thrpt.↑ | Acc.↑ | Thrpt.↑ | Acc.↑ | Thrpt.↑ | Acc.↑ | Thrpt.↑ |
| Dream 7B (Random, 2 token/step) | 38.02 | 15.42 | 14.57 | 30.59 | 35.13 | 12.34 | 34.00 | 5.50 |
| Dream 7B (Entropy, 4 token/step) | 52.27 | 47.37 | 11.73 | 47.81 | 32.32 | 30.60 | 49.13 | 15.37 |
| Dream 7B (Entropy, 2 token/step) | 72.00 | 26.25 | 27.91 | 29.13 | 47.56 | 17.44 | 55.93 | 8.26 |
| Qwen2.5 7B (Autoregressive) | 95.53 | 49.26 | 76.28 | 46.50 | 83.54 | 45.83 | 75.87 | 47.09 |
| PoE-Bridge w/o IS | 95.20 | 104.49 | 73.86 | 99.49 | 80.69 | 84.82 | 72.20 | 79.65 |
| PoE-Bridge | 95.30 | 100.71 | 74.42 | 94.94 | 79.47 | 76.13 | 73.20 | 72.10 |

Therefore, speculative rejection sampling from tempered PoE distributions can be implemented efficiently by simply combining tempered DLM proposal logits with tempered AR verification logits at each position.

Since the candidates are now drawn from different proposals $q_{\tau_k}$, we use a multiple-importance resampling rule (Veach & Guibas, 1995). Let $\tau_\star$ denote the target temperature (we use $\tau_\star = \tau_{\text{low}}$ by default). For a candidate $\hat{\mathbf{x}}_{c:c+a_k-1}^{(k)} \sim q_{\tau_k}$, we assign it an importance weight

$$w_k \propto \frac{p_{\text{AR}}(\hat{\mathbf{x}}_{c:c+a_k-1}^{(k)} \mid \mathbf{x}_{<c})^{1/\tau_\star}/Z_{\text{AR}}(\tau_\star)}{p_{\text{PoE}}(\hat{\mathbf{x}}_{c:c+a_k-1}^{(k)} \mid \mathbf{x}_{<c})^{1/\tau_k}/Z_{\text{PoE}}(\tau_k)}, \quad (16)$$

where $Z_{\text{AR}}$ and $Z_{\text{PoE}}$ denote the temperature-dependent normalizing constants of the tempered AR target and tempered PoE proposal, respectively. In principle, $Z_{\text{AR}}(\tau_\star)$ can be omitted because it is shared across all candidates and therefore cancels when normalizing the raw weights. In contrast, $Z_{\text{PoE}}(\tau_k)$ depends on the proposal temperature and therefore differs across candidates. Omitting it introduces bias by rescaling the relative contributions of samples drawn from different temperatures. Nevertheless, for simplicity, we omit all these normalizers in practice, which empirically performs well. Overall, mixed-temperature sampling improves candidate diversity and increases the effectiveness of importance resampling under a fixed compute budget, as we demonstrate empirically in Section 5.3.

### 4.4. Elastic Rejection Window

At inference time, diffusion LMs are typically queried with a sufficiently long masked suffix to avoid distribution shift relative to training, especially near boundary symbols such as the end-of-sequence (EOS) token (Israel et al., 2025). Concretely, given a maximum length $n$, the model often takes as input a prefix ending at position $c$ together with a large masked continuation length $n - c$. However, our rejection-based verification usually accepts only a short prefix of the proposed continuation. Generating and verifying an entire long suffix is therefore wasteful and reduces the

number of parallel candidates $K$ we can afford under a fixed compute budget.

To reduce wasted computation, we introduce an *Elastic Rejection Window* that restricts parallel proposal and verification to the next $V$ positions at a time. At prefix index $c$, we sample proposals in parallel only for the window $\mathbf{x}_{c:c+V-1}$—rather than the entire suffix $\mathbf{x}_{c:n}$—from $\boldsymbol{\mu}_\theta(\cdot \mid \mathbf{x}_{<c})$ and evaluate the corresponding AR likelihoods $p_{\text{AR}}(\mathbf{x}_{c:c+V-1} \mid \mathbf{x}_{<c})$ for rejection. If all $V$ tokens are accepted, we advance $c \leftarrow c + V$ and reuse $\boldsymbol{\mu}_\theta$'s output logits in the next window; otherwise, we restart from the first rejected position. Further details are provided in Algorithm 1. Importantly, this windowing does not change the output distribution: it only truncates computation that would be discarded after a rejection, while preserving the same acceptance and correction rules.

In practice, the window size $V$ trades off parallelism against wasted verification: larger windows increase parallelism but may verify many ultimately rejected tokens, while smaller windows reduce waste at the cost of more sequential steps; we analyze this tradeoff in detail in Section 5.3.

## 5. Experiments

### 5.1. Experiment Setup

**Benchmarks.** Our goal is to evaluate whether PoE-Bridge can generate faithful samples following the distribution of a strong autoregressive (AR) model while taking advantage of the efficient parallel generation capability of a diffusion language model (DLM). We therefore focus on challenging mathematical reasoning and coding benchmarks, where existing DLMs are known to substantially lag behind AR models. Specifically, we consider the math reasoning tasks GSM8K (Cobbe et al., 2021) and MATH (Hendrycks et al., 2021), as well as the coding benchmarks HumanEval (Chen, 2021) and MBPP (Austin et al., 2021b).

**Implementation details.** We select models from the Dream (Ye et al., 2025) and Qwen2.5 (Team, 2024) fam-

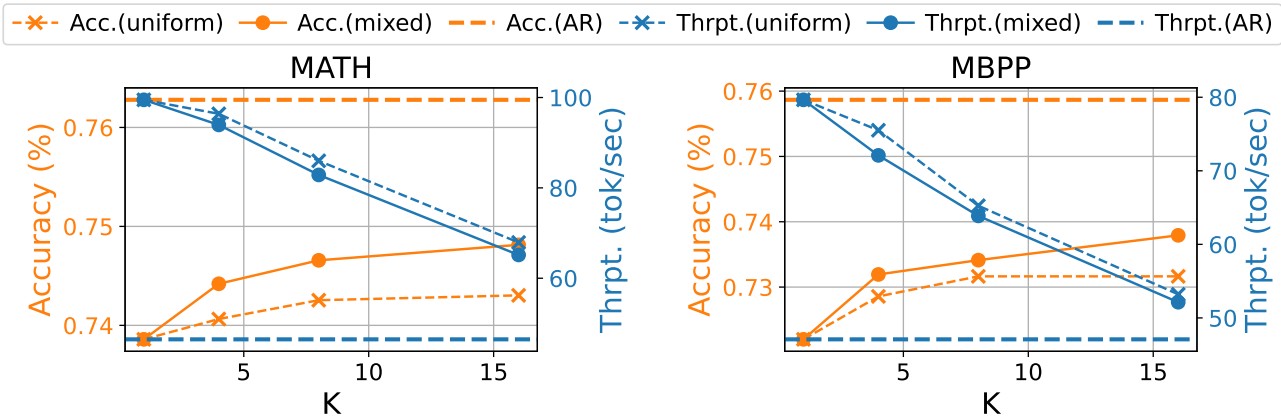

*Figure 2.* Effect of increasing the number of parallel candidates $K$ under uniform- and mixed-temperature sampling. Mixed-temperature sampling enables consistent accuracy improvements with increasing $K$, whereas uniform-temperature sampling yields early-plateau returns.

ilies, as they share the same tokenizer and vocabulary. Throughout all experiments, we use Dream-7B-Instruct as the DLM proposal. For task-specific AR experts, we employ Qwen2.5-Math-7B-Instruct (Yang et al., 2024) on mathematical reasoning benchmarks and Qwen2.5-Coder-7B-Instruct (Hui et al., 2024) on coding tasks.

All models are evaluated in BF16 precision on a single NVIDIA A100 GPU (80GB). We use the standard LM Evaluation Harness (Gao et al., 2023), with minor modifications to input instructions and answer parsing for improved accuracy. Unless otherwise specified, we use PoE weight $w = 0.3$, parallel candidate size $K = 4$, and an Elastic Rejection Window of size $V = 32$. We analyze the impact of these hyperparameters in Section 5.3. Additional implementation details are provided in Section A.

### 5.2. Main Results

We start by showing that PoE-Bridge substantially improves *both* generation quality and decoding speed relative to standard DLM decoding, while closely matching the accuracy of strong AR Target. As shown in Table 1, across all benchmarks, PoE-Bridge achieves significantly higher accuracy with up to $5\times$ speedup than the standard entropy-based DLM decoding. Notably, our approach also recovers over $95\%$ of AR accuracy while achieving $2\times$ the throughput of AR decoding. This demonstrates that our approach breaks the conventional quality–efficiency tradeoff of diffusion language models, achieving AR-level faithfulness without sacrificing decoding speed.

The throughput gain stems from the fact that PoE-Bridge accepts multiple tokens per DLM forward pass. In contrast to standard DLM decoding, which typically commits only one or two tokens per step to avoid significant quality degradation, our method enables longer accepted token blocks,

leading to significantly higher effective throughput. As a result, the additional AR forward pass does not negate the efficiency gains from parallel DLM generation.

We further analyze the contribution of the importance sampling stage in our decoding algorithm. Removing the importance sampling step (i.e., setting K=1, corresponding to PoE-Bridge w/o IS in Table 1) yields a small throughput improvement and slightly degrades accuracy. This suggests that the PoE-based rejection step alone already pushes samples toward the AR target, but importance sampling over multiple candidates is necessary to further recover high-fidelity generations. The limited throughput gain in this setting is expected: with our Elastic Rejection Window, both candidate sampling and AR likelihood evaluation remain highly parallelized on modern GPUs, so reducing K provides only marginal savings.

We also provide qualitative examples in Section C to illustrate the behavioral differences between PoE-Bridge and baseline decoding strategies.

### 5.3. Ablation Studies

**Mixed-temperature sampling enables effective scaling with $K$.** We study how increasing the number of parallel candidates $K$ in the importance sampling phase affects convergence toward the target AR model, and show that meaningful scaling emerges only when candidates are drawn from a mixed set of temperatures. We conduct this ablation on the harder MATH and MBPP benchmarks, where the larger gap between the DLM and AR target makes performance gains more apparent. Across all settings, we fix $w = 0.3$.

All results are summarized in Figure 2. Under uniform-temperature sampling, where all candidates are drawn from the target temperature, increasing $K$ yields little benefit. As shown by the dotted curves, accuracy improves only

*Table 2.* Effect of PoE weight $w$ on accuracy, throughput, accepted tokens per DLM forward pass (Tok/Step), and the percentage of decoding steps in which all drafted tokens are rejected (% All Reject). The setting $w = 0.0$ corresponds to direct speculative decoding from the DLM proposal and the AR target, without an intermediate PoE bridge distribution.

| Metric / $w$ | 0.0 | 0.3 | 0.5 | 0.7 | 0.9 |
|---|---|---|---|---|---|
| Acc.↑ | 81.27 | 80.69 | 74.39 | 64.63 | 30.49 |
| Thrpt.↑ | 56.55 | 84.82 | 85.13 | 104.27 | 168.35 |
| Tok/Step↑ | 4.91 | 8.30 | 8.32 | 9.25 | 15.61 |
| % All Reject↓ | 17.15 | 4.53 | 2.73 | 1.88 | 0.33 |

*Table 3.* Effect of elastic rejection window size $V$ on throughput. When $V = \infty$, the Elastic Rejection Window is disabled, and all logits produced by each DLM forward are sampled and verified.

| $K$ / $V$ | 8 | 16 | 32 | 64 | $\infty$ |
|---|---|---|---|---|---|
| $K = 1$ | 66.22 | 65.79 | 99.49 | 91.37 | 83.91 |
| $K = 4$ | 80.96 | 94.99 | 94.94 | 71.43 | 39.46 |

marginally as $K$ grows, while throughput steadily degrades once the additional sampling and verification costs exceed available hardware parallelism. This indicates that simply increasing the number of parallel samples is insufficient to close the gap between the DLM and the target AR model.

In contrast, mixed-temperature sampling exhibits clear and consistent scaling behavior, as $K$ increases, accuracy steadily improves and moves closer to the AR target. Within the range of $K$ supported by our compute budget, mixed-temperature sampling closes at least one-third of the remaining gap between the PoE candidates and the AR target.

In Section B, we report how additional statistics vary as $K$ to provide a more detailed view on the generative behaviors.

**Quality–efficiency tradeoff *w.r.t* the PoE weight $w$.** Next, we study the PoE weight w, which controls the PoE's proximity to the DLM proposal and thus the aggressiveness of rejection. Larger $w$ moves the bridge toward the proposal, increasing acceptance rates and throughput. We conduct this ablation on HumanEval with $K = 1$ to better expose quality differences. As shown in Table 2, increasing $w$ consistently improves throughput, with longer accepted prefixes and fewer decoding steps in which all proposal tokens are rejected. However, the efficiency gain comes at the cost of degraded accuracy. For larger $w$, quality degradation accelerates and outweighs the marginal throughput gains, indicating an unfavorable configuration. Overall, we find that $w = 0.3$ provides the best quality–efficiency tradeoff, recovering high accuracy while maintaining substantial throughput gains.

**Elastic Verification Window.** Finally, we analyze the Elastic Verification Window, which balances wasted verification against parallelism. We run this ablation on MATH, where longer responses amplify verification overhead. We use the default $w = 0.3$ and vary $K = 1, 4$ to study the effect of $V$ under different sampling budgets. As shown in Table 3, throughput degraded when $V$ is too small or too large. Small $V$ fragments acceptances across sequential iterations, reducing parallelism. This effect is stronger when

$K$ is small, as the GPU still has capacity to accommodate larger $V$. When $V$ is too large, verification spans become long enough to exceed available hardware parallelism, and meanwhile, more computation is spent verifying tokens that are ultimately rejected. Large $V$'s impact is more evident when $K$ is large, where hardware parallelisms are already saturated. Overall, the elastic window is what makes importance sampling practical in our setting: by adapting verification spans to realized acceptances and hardware capacity, it avoids both excessive wasted verification and excessive sequential fragmentation.

## 6. Limitations

Our experiments only consider ARLM–DLM pairs with a shared tokenizer. Extending PoE-Bridge to mismatched tokenizers is feasible in principle using token-alignment techniques from speculative decoding (Timor et al., 2025), but we leave this direction for future work. Our current implementation is evaluated only in the single-GPU, batch-size-1 setting, with both models resident during decoding. This increases memory pressure and leaves multi-query serving unstudied. Concurrent single-model approaches (Kumar et al., 2026; Liu et al., 2025) may help alleviate this overhead, and we leave such integration to future work. PoE-Bridge is not strictly lossless under a finite importance-sampling budget. However, the observed degradation is small in practice and decreases as $K$ scales up, as shown in Figure 2.

## 7. Conclusion

In this work, we study the problem of achieving autoregressive-faithful text generation without sacrificing parallel decoding efficiency. We introduce PoE-Bridge, a decoding framework that enables diffusion language models to produce high-quality generations while preserving parallelism. By reducing proposal–target mismatch through an intermediate bridge distribution, PoE-Bridge makes Monte Carlo correction effective under limited compute budgets. We further introduce practical techniques to enable efficient and scalable implementation. Experiments demonstrate that PoE-Bridge approaches AR-level accuracy while delivering substantial decoding speedups, highlighting its potential as a practical solution for high-quality parallel text generation.

# Acknowledgements

This work was supported by ARO (W911NF-21-1-0125), ONR (N00014-23-1-2159), the CZ Biohub, NSF under Nos. CCF-1918940 (Expeditions), DMS-2327709 (IHBEM), IIS-2403318 (III); NIH under No. 1U24NS146314-01, Stanford Data Applications Initiative, Wu Tsai Neurosciences Institute, Stanford Institute for Human-Centered AI, Chan Zuckerberg Initiative, Amazon, Genentech, SAP, and SCBX,

# Impact Statement

We confirm that this work complies with the ICML Publication Ethics, and we have carefully considered potential ethical implications related to the development and use of PoE-Bridge, a decoding framework for diffusion language models.

PoE-Bridge is a general-purpose inference-time method designed to improve the efficiency and faithfulness of text generation by combining diffusion language models with autoregressive verification. It does not introduce new training data, modify model parameters, or rely on sensitive or private datasets. All models and benchmarks used in our experiments are publicly available and well-documented.

We also note that PoE-Bridge operates entirely at inference time and does not inherently mitigate or exacerbate privacy risks beyond those already present in the underlying language models. Users should therefore continue to evaluate privacy and data leakage risks according to the domains in which the base models are applied, and ensure compliance with relevant regulations and best practices.

We are committed to transparency and reproducibility. All experimental settings, assumptions, and implementation details are clearly documented (see Section A), and we release our code to facilitate reproducibility and future research. No conflicts of interest, financial or otherwise, influenced the development or presentation of this work.

Based on these considerations, we do not anticipate violations of the ICML Publication Ethics arising directly from PoE-Bridge. However, we stress that the method should not be used for malicious purposes, and that responsible deployment remains the responsibility of system designers and users.

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

# A. Implementation Details

**Generation length and termination.** Allowing models to complete their full answers is crucial for accurate evaluation. For all DLM-based methods, including PoE-Bridge, we cap the maximum number of newly generated tokens at 2,048, which we found sufficient for the vast majority of examples. For the autoregressive (AR) baseline, we enforce no additional limits on generation length besides the default context window limit of 16,384 tokens. All methods terminate generation early when the end-of-sequence (EOS) token is produced. Throughput is computed as the number of generated tokens before EOS divided by the total wall-clock decoding time.

**DLM forward configuration.** When running DLM forwards in PoE-Bridge, we pass in the current generated tokens along with 256 trailing mask tokens, instead of all mask tokens up to the maximum generation length. This avoids costly attention computation over the full context, significantly reducing computation, while introducing only a minor distribution shift in practice. For diffusion-based baseline decoding methods, we use a block size of 256 to match this configuration and ensure a fair comparison.

**Evaluation protocol.** All reported results are averaged over three runs with different random seeds, as we observe moderate variance across runs. We evaluate the full datasets for GSM8K, HumanEval, and MBPP. For MATH, we evaluate the first 215 problem sets, as the full dataset (1,500 problems) is approximately five times larger than GSM8K and would otherwise dominate evaluation cost.

**Prompts and answer parsing.** We evaluate instruction-tuned models in a zero-shot setting. The default LM Evaluation Harness (lm_eval) pipelines are designed for base models and rely on few-shot exemplars to enforce answer formats. Directly applying these pipelines to instruction-tuned models leads to inaccurate evaluation, as the model may place the final answer arbitrarily within the output. To address this, we make minor modifications to both the prompts and answer parsers.

For GSM8K and MATH, we use task-specific prompts that explicitly instruct the model to present its final answer in \boxed{} format at the end of the response; for MATH, we additionally specify the nature of the reasoning task. During evaluation, we parse the content of the final boxed expression. If no boxed answer is found for GSM8K, we fall back to extracting the last numerical value in the generation, consistent with the task's default evaluation protocol. For MATH, only the boxed answer is used.

For the coding benchmarks HumanEval and MBPP, we use the default "You are a helpful assistant" prompt and the standard answer parsers provided by the evaluation framework.

**Detailed temperature schedule.** To improve candidate diversity under a fixed parallel budget $K$, we draw candidates from a set of tempered PoE distributions with different temperatures. Specifically, we assign each candidate $k \in \{1, \ldots, K\}$ a temperature

$$\tau_k \leftarrow \tau_{\text{low}} + \frac{k-1}{K-1}\big(\tau_{\text{high}} - \tau_{\text{low}}\big) \tag{17}$$

which linearly interpolates between a low temperature $\tau_{\text{low}}$ and a high temperature $\tau_{\text{high}}$. Candidate $k$ is then sampled from the tempered PoE distribution

$$\begin{aligned} q_{\tau_k}(\mathbf{x}_i \mid \mathbf{x}_{<i}) &\propto \tilde{p}_D(\mathbf{x}_i \mid \mathbf{x}_{<i})^{w/\tau_k} \, p_{\text{AR}}(\mathbf{x}_i \mid \mathbf{x}_{<i})^{(1-w)/\tau_k} \\ &= \Big(\tilde{p}_D(\mathbf{x}_i \mid \mathbf{x}_{<i})^{1/\tau_k}\Big)^w \Big(p_{\text{AR}}(\mathbf{x}_i \mid \mathbf{x}_{<i})^{1/\tau_k}\Big)^{1-w}. \end{aligned} \tag{18}$$

Equivalently, this corresponds to first applying temperature $\tau_k$ to both the DLM proposal and the AR expert, and then combining them through the PoE. We rely on this fact in our implementation.

In all experiments, we use $\tau_{\text{low}} = 0.2$ and $\tau_{\text{high}} = 0.7$. During importance resampling, the target temperature $\tau_\star$ is set to $\tau_{\text{low}}$, so that all candidates are corrected toward the same low-temperature target distribution.

# B. Additional Results

**Additional statistics for the ablation study on the scaling effect of $K$.** In Figures 3 and 4, we show two additional statistics as the number of parallel candidates $K$ increases: the generation length and the average number of tokens accepted per DLM forward pass. Both quantities generally increase with larger $K$. Since the target AR model typically produces answers with longer and more detailed reasoning chains, this trend provides further evidence that increasing $K$ pushes the generated samples closer to the target distribution.

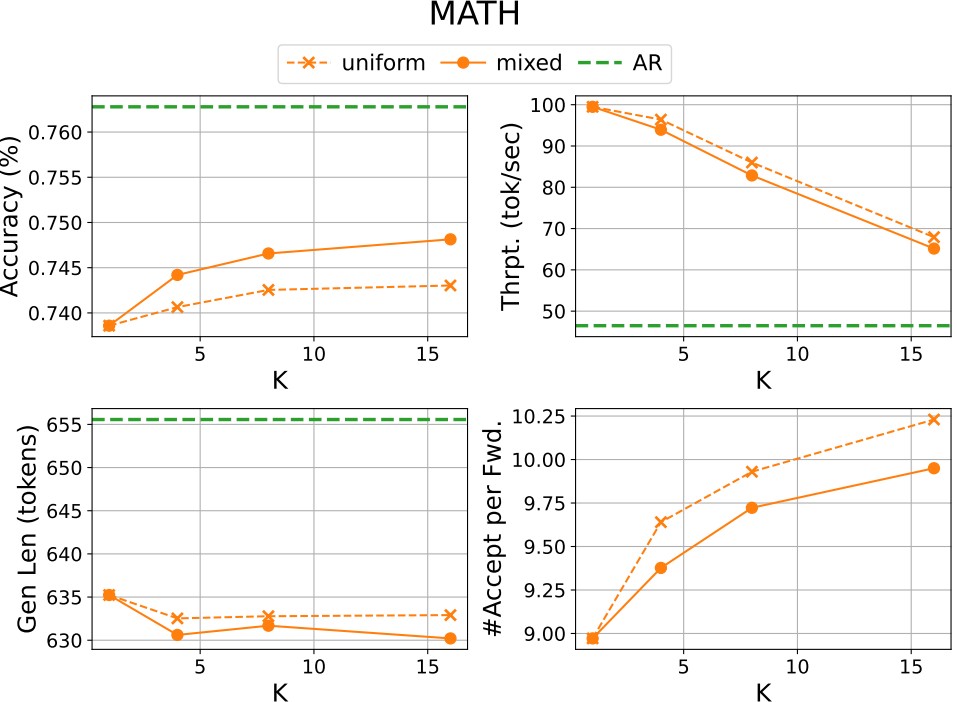

*Figure 3.* Additional statistics for the ablation study on the scaling effect of $K$, conducted on MATH. Since the AR decoding baseline does not have the corresponding statistics for the #Accept per Fwd. statistics, we omit it in that subplot.

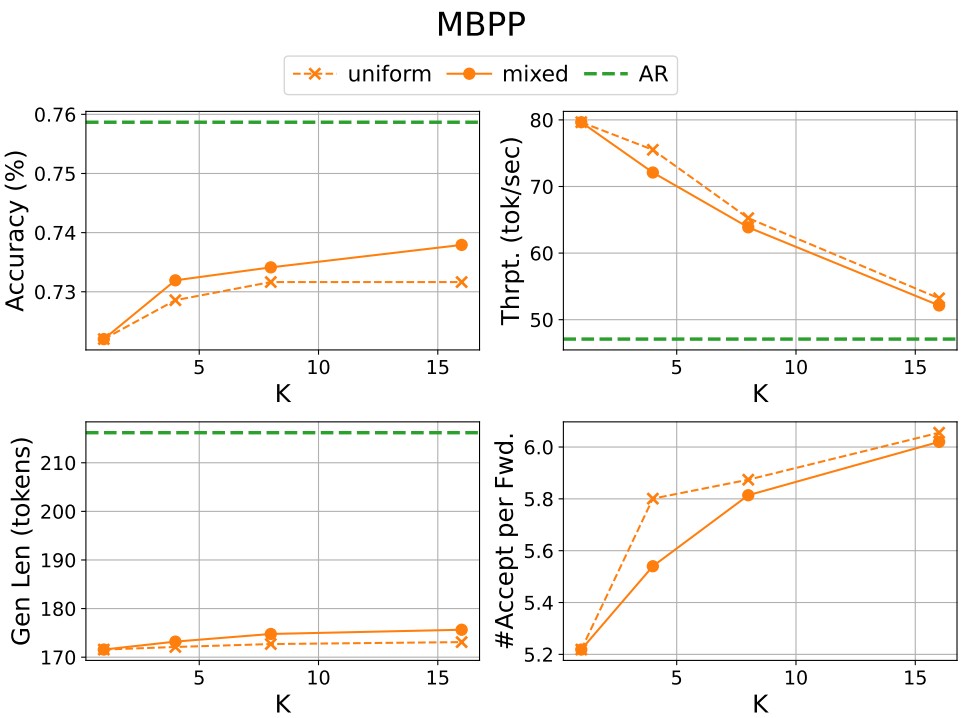

*Figure 4.* Additional statistics for the ablation study on the scaling effect of $K$, conducted on MBPP. Since the AR decoding baseline does not have the corresponding statistics for the #Accept per Fwd. statistics, we omit it in that subplot.

## C. Qualitative Examples

Below, we show responses generated by PoE-Bridge alongside those from the DLM proposal and the AR target on math reasoning tasks. The examples illustrate that PoE-Bridge closely mimics the style of the post-trained AR target. In particular, both PoE-Bridge and the AR target produce substantially longer reasoning chains than the original DLM proposal. This qualitative behavior further suggests that PoE-Bridge effectively recovers the target distribution.

> **Question:**
>
> Kyle bought last year's best-selling book for $19.50. This is with a 25% discount from the original price. What was the original price of the book?

> **Standard DLM Answer:**
>
> To find the original price of the book, we need to divide the discounted price by the percentage of the original price that remains. The discounted price is $19.50, which is 87.5% of the original price. So the original price is $\boxed{22.50}$.

> **PoE-Bridge Answer:**
>
> To find the original price of the book, we determine the price before the 25% discount was applied. Let the original price be denoted by $P$.
> Since Kyle paid $19.50 after a 25% discount, we have
> $$19.50 = P - 0.25P.$$
> Simplifying the right-hand side,
> $$19.50 = 0.75P.$$
> Solving for $P$ by dividing both sides by 0.75,
> $$P = \frac{19.50}{0.75} = 26.$$
> Therefore, the original price of the book was
> $$\boxed{26}.$$

> **AR Answer:**
>
> To find the original price of the book, we determine the price before the 25% discount was applied. Let the original price be denoted by $P$.
> Since Kyle bought the book for $19.50 after a 25% discount, he paid 75% of the original price. Therefore, we can write
> $$0.75P = 19.50.$$
> Solving for $P$ by dividing both sides by 0.75,
> $$P = \frac{19.50}{0.75}.$$
> Evaluating the expression,
> $$P = 26.$$
> Thus, the original price of the book was
> $$\boxed{26}.$$

**Question:**

Evaluate $i^5 + i^{-25} + i^{45}$.

**Standard DLM Answer:**

To evaluate $i^5 + i^{-25} + i^{45}$, we need to understand the properties of $i$, the imaginary unit, which is defined as $i = \sqrt{-1}$. The powers of $i$ cycle every four terms: $i^1 = i$, $i^2 = -1$, $i^3 = -i$, $i^4 = 1$, and then it repeats.
1. $i^5$ is simply $i$.
2. $i^{-25}$ can be simplified by recognizing that $i^{-2} = -1$, so $i^{-25} = (i^{-2})^{12}$
$cdoti^{-1} = (-1)^{12}$
$cdoti^{-1} = -1$.
3. $i^{45}$ can be simplified by recognizing that $i^4 = 1$, so $i^{45} = (i^4)^{11}$
$cdoti^1 = 1^{11}$
$cdoti = i$.

Adding these together, we get $i + (-1) + i = 1 - 1$.

$\boxed{0}$.

**PoE-Bridge Answer:**

To evaluate the expression $i^5 + i^{-25} + i^{45}$, we use the properties of the imaginary unit $i$, where

$$i = \sqrt{-1}.$$

The powers of $i$ cycle every four terms:

$$i^1 = i, \quad i^2 = -1, \quad i^3 = -i, \quad i^4 = 1,$$

and this pattern repeats for higher powers.
We simplify each term individually.

1. **Simplify $i^5$:**
$$i^5 = i^{4+1} = i^4 \cdot i = 1 \cdot i = i.$$

2. **Simplify $i^{-25}$:** Using the identity $i^{-n} = \frac{1}{i^n}$, we first compute
$$i^{25} = i^{4\cdot6+1} = (i^4)^6 \cdot i = 1^6 \cdot i = i.$$

   Thus,
$$i^{-25} = \frac{1}{i}.$$

   To rationalize the denominator,
$$\frac{1}{i} = \frac{i}{i^2} = \frac{i}{-1} = -i.$$

3. **Simplify $i^{45}$:**
$$i^{45} = i^{4\cdot11+1} = (i^4)^{11} \cdot i = 1^{11} \cdot i = i.$$

Substituting these results back into the original expression,

$$i^5 + i^{-25} + i^{45} = i + (-i) + i = i.$$

$\boxed{i}$

---

**AR Answer:**

To evaluate $i^5 + i^{-25} + i^{45}$, we start by recalling the properties of the imaginary unit $i$, where $i = \sqrt{-1}$ and the powers of $i$ cycle every four terms: $i^1 = i$, $i^2 = -1$, $i^3 = -i$, and $i^4 = 1$. This cycle repeats for higher powers of $i$.

First, we simplify $i^5$:
$$i^5 = i^{4+1} = i^4 \cdot i^1 = 1 \cdot i = i.$$

Next, we simplify $i^{-25}$. Since the powers of $i$ cycle every four terms, we can find the equivalent positive exponent by computing $-25 \bmod 4$:

$$-25 \div 4 = -6 \text{ remainder } -1 \quad \text{or} \quad -25 \equiv -1 \pmod{4}.$$

Since $-1 \equiv 3 \pmod{4}$, we have:
$$i^{-25} = i^3 = -i.$$

Then, we simplify $i^{45}$:
$$i^{45} = i^{44+1} = i^{44} \cdot i^1 = (i^4)^{11} \cdot i = 1^{11} \cdot i = 1 \cdot i = i.$$

Now, we add the simplified terms together:

$$i^5 + i^{-25} + i^{45} = i + (-i) + i = i - i + i = i.$$

Therefore, the final answer is:
$$\boxed{i}.$$

