# OpenReview forum: "Diffusion Language Model Parallel Decoding via Product-of-Experts Bridge"
_ICML.cc/2026/Conference — ICML 2026 regular_

### Official Review · Reviewer_Pma8 · 2026-03-09

**Soundness:** 3
**Presentation:** 3
**Significance:** 3
**Originality:** 3
**Overall Recommendation:** 4
**Confidence:** 4

**Summary:**

The paper proposes inference acceleration of AR LM using multi-token proposals from DLM,  where the AR LM acts as the target density and DLM as a proposal. A key challenge in such as setup is the mismatch between the two densities, which leads to high rejection rates. To address this issue, the paper proposes to use an interpolation of the two densities as the target density. Additionally, it proposes two stage sampling procedure that uses rejection like step and importance sampling step at span level. The importance sampling uses candidates that are obtained by sampling using different temperatures from the DLM. The experiments show approximately 2x improvement in throughput w.r.t AR LM with minimal loss in accuracy. Key ablations also show the impact of the proposed rejection-with-correction step and importance sampling step.

**Compliance With Llm Reviewing Policy:**

Affirmed.

**Ethical Review Flag:**

Flag this paper for an ethics review.

**Final Justification:**

Apologies for the delay in my final justification. The authors have answered my questions satisfactorily. As mentioned in my review, I believe that there some limitations of a two-model approach that are better addressed by more lightweight single model approaches. That said the paper presents an interesting method to reduce the density mismatch between the DLM and ALM and provides good empirical results and would be useful for the community. Therefore, I'd like to maintain my original positive recommendation.

**Key Questions For Authors:**

1. How as the throughput measured exactly? Does it include non-GPU operations?
2. The experiments use math and code tasks. However, how would the proposed method work for more open-ended generation tasks? Would it require significantly different hyperparemter settings?
3. There is memory cost of considering multiple candidates and loading two different models in the GPU memory. Can you comment on how this affects the amount of queries the setup can handle per batch, and also discuss how the two models are handled (e.g. are they on the same GPU or different GPUs).
4. I understand that the proposed approach is purely an inference time strategy but it might still be beneficial to discuss some concurrent single model approaches that try to address the same problem [1].

[1] Liu, Jingyu, et al. "Tidar: Think in diffusion, talk in autoregression." arXiv preprint arXiv:2511.08923 (2025).

**Limitations:**

Please mention and discuss some limitations of the proposed approach, for instance the engineering and memory overhead of managing two models.

**Strengths And Weaknesses:**

**Soundness**: The experiments support the central claims of the paper well.

**Presentation**: The paper is well written and easy to follow.

**Significance**: Accelerating inference of LLMs is an important problem, and the paper proposes a promising approach with good empirical results, making the paper of great interested for the community. I'm not sure how straightforward this would but having experiments (or at least a discussion) where the AR LM and DLM use different tokenizers could be interesting and a more general setting.

**Originality**: The proposed approach is closely related to many existing inference acceleration methods that use two models. However, the use of interpolated target density and the specifics of rejection-with-correction and multi-temperature importance sampling are novel. Moreover, the ablations provide useful insights as well.

---

> ### Author Rebuttal · Authors · 2026-03-31
>
> We appreciate the positive feedback and address the points below.
>
> >**[W1] Applicability to ARLM-DLM pairs with different tokenizers**
>
> Conceptually, PoE-Bridge only requires aligned token-level probabilities from the DLM and ARLM under a common factorization of the sequence to compute the mixture density. Identical tokenizers are not required, although a shared tokenizer is the simplest case.
>
> In the general case where tokenizers differ arbitrarily, recent work [1] in speculative decoding provides a solution: it decodes draft tokens to text, re-tokenizes with the target tokenizer, and aligns via longest-overlap matching. The same idea extends directly to PoE-Bridge, enabling mixture-density computation and rejection sampling in the ARLM token space.
>
> >**[Q1] How is the throughput measured exactly**
>
> As stated in Appendix A, the throughput is computed as the ratio between a) the number of generated tokens (excluding the input prompt) and b) the total wall-clock decoding time until EOS. The timing includes the full decoding process, but excludes one-time overheads, such as model loading and tokenization. In practice, it is dominated by GPU operations, including AR/DLM forward passes and sampling.
>
> >**[Q2] Performance on open-ended generation tasks?**
>
> PoE-Bridge is task-agnostic in formulation, so it is not specific to math or code tasks. Its performance instead depends on two factors. First, the achievable speedup depends on the distribution gap between the DLM and the AR target; this is a general challenge for speculative-style decoding methods, although PoE-Bridge is specifically designed to mitigate it by reducing rejection. Second, the output quality is limited by the quality of the AR target model, since PoE-Bridge guides samples towards that target.
>
> To test generalization beyond math and code, we additionally evaluate on the science QA benchmark GPQA[2]. We intentionally choose a mismatched AR target, Qwen-Math, which is post-trained for a different domain and in fact performs worse than the base DLM on this task. We use the same hyperparameters as in the main experiments.
>
> ||Acc.|Thrpt.|
> |---|:-:|:-:|
> |Dream 7B (entropy, 2tok/step)|28.20|17.64|
> |Qwen2.5 7B|15.92|46.88|
> |PoE-Bridge (K=1, w=0.3)|17.86|57.86|
> |PoE-Bridge (K=4, w=0.3)|18.75|59.44|
>
> These results show that PoE-Bridge still tracks the AR target even in this mismatched setting, while maintaining higher throughput than both baselines.
>
> We do not expect fundamentally different hyperparameters to be required for more open-ended tasks, although the best operating point may shift depending on the proposal–target gap and the desired quality–throughput tradeoff.
>
> We will also add qualitative open-ended examples in the revised draft.
>
> >**[Q3] Memory management and cost**
>
> All our experiments are run on a single GPU with query batch size 1. Both the DLM and ARLM reside on the same GPU throughout decoding. The main memory overhead comes from storing both model weights, and the cost of maintaining $K$ importance-sampling candidates. We have not yet studied the multi-query setting. While feasible in principle, it requires more complex control flow and padding, similar to speculative decoding.
>
> In our setting, the more important issue with larger $K$ is compute, not memory. Although the candidate evaluations induced by $K$ are highly parallelizable, on a single GPU, they can still reduce throughput substantially if handled naively, once the workload exceeds the available parallelism. This is precisely why we designed the *elastic rejection window*: it mitigates this issue by avoiding likely wasted computation, making moderate values such as $K=4$ practical in our experiments. Under a multi-GPU or more specialized architecture, this candidate computation would likely be even easier to amortize, as it is inherently parallelizable.
>
> >**[Q4] Comparison with TiDAR**
>
> TiDAR[3] is a relevant concurrent work, but it differs fundamentally from PoE-Bridge in scope and mechanism. We thus view the two as largely orthogonal and potentially composable. Due to space limits, please see Reviewer `BVgP` **[W1.1]** for details.
>
> >**Limitations**
>
> We summarize the main limitations here and will add them in the revision.
>
> - We only evaluate ARLM–DLM pairs with a shared tokenizer; extending to mismatched tokenizers is feasible in principle using token-matching techniques, but we leave for future work.
> - We only evaluate the single-GPU, batch-size-1 setting, with both models resident during decoding. This increases memory pressure and leaves multi-query serving unstudied. Single-model approaches may alleviate this overhead, and we leave such integration to future work.
> - PoE-Bridge is not strictly lossless under a finite importance-sampling budget. But degradation is small in practice and decreases as $K$ scales up (as shown in Figure 1).
>
> [1] Timor et al. “Accelerating LLM Inference…” ICML, 2025.
>
> [2] Rein et al. “GPQA…” COLM, 2024.
>
> [3] Liu et al. “TiDAR…” arXiv, 2025.

---

> > ### Author Rebuttal · Reviewer_Pma8 · 2026-04-02
> >
> > Thank you for answering my questions.

---

### Official Review · Reviewer_TLW6 · 2026-03-13

**Soundness:** 3
**Presentation:** 3
**Significance:** 2
**Originality:** 3
**Overall Recommendation:** 4
**Confidence:** 4

**Summary:**

This paper presents Product-of-Experts Bridge, utilizing a shared distribution between an autoregressive model and a diffusion language model for rejection sampling and importance sampling. The authors have additionally proposed mixed-termperature importance sampling and elastic rejection window to ensure the generation quality and efficiency. The final experiment results show that PoE bridge can improve the generation quality and output throughput by a large margin compared to a DLM using entropy-based sampling.

**Compliance With Llm Reviewing Policy:**

Affirmed.

**Final Justification:**

Overall, the paper presents PoE bridge, which is a novel method to bridge the gap between the AR and DLM distributions. However, as a method similar to speculative decoding, more thorough comparison with existing methods such as EAGLE-3 and DFlash in terms of the model performance and system runtime metrics is required.

**Key Questions For Authors:**

1. In In Naive Approaches (Line 184), it is stated that "In practice, however, the huge discrepancy between pD and pAR leads to frequent rejection, so the accepted prefix is typically short." Can you provide some evidence for this claim as this is the fundamental motivation?
2. In Table 1, I am wondering why Qwen2.5 7B performs better both in terms accuracy and throughput. As DLMs are well-known for the parallel decoding capability, I would expect that Dream 7B has a higher output throughput.
3. If I understand correctly, this work looks more like a combination of speculative decoding (rejection sampling) and importance sampling. The difference in the speculative decoding part is that it uses the bridge distribution instead. In this sense, would it be more reasonable to include speculative decoding methods in the result comparison?
4. The Table 1 only presents comparison between PoE and entropy-based DLM sampling, there are other acceleration methods for DLMs, I am wondering if it could be better to include other baselines?

**Limitations:**

1. In PoE bridge, both the AR and DLM models must be present during infernece, this adds more pressure to the GPU memory during serving.

**Strengths And Weaknesses:**

Strenght:
1. The paper is generally easy to follow with clear logic flow.
2. The proposal of PoE bridge is novel as the authors find the fundamental distribution mismatch between the AR and DLM models and proposed PoE bridge with solid maths foundation.
3. The results look promising as it is faster than the base DLM by almost 2x and delivers generation quality close to the AR model.

Weakness:
1. In terms of readability, having some figures to show the architecture of PoE bridge would help readers better understand its workflow.
2. Other weaknesses can be found in my questions below.

---

> ### Author Rebuttal · Authors · 2026-03-31
>
> We thank the reviewer for the positive feedback and address the questions below.
>
> >**[W1] Figure to illustrate the method workflow**
>
> We agree that a workflow figure for our method would improve readability and make it easier to follow. We will include one in the revised draft.
>
> >**[Q2] Why does the ARLM outperform baseline DLM decoding in both accuracy and throughput?**
>
> This is partly due to our setup: the AR baselines in Table 1 are task-specialized post-trained models (e.g., Qwen-Math / Qwen-Coder), whereas the DLM is a base model. We intentionally use a challenging proposal–target pair so that the benefit of PoE-Bridge is clearly visible.
>
> More generally, even when comparing against base AR models, current open-source DLMs with standard decoding methods still do not offer a better quality–throughput tradeoff: their parallel speedup often comes at a substantial quality cost, and pushing them toward AR-level quality tends to make decoding nearly sequential, as also reflected in APD [1]. We believe this limitation is fundamentally tied to the conditional independence assumptions used by DLMs to enable parallel decoding.
>
> >**[Q1, Q3] Evidence for frequent rejection, and comparison with speculative-decoding baselines**
>
> To support the claim and provide additional baseline comparisons, we present additional experiments with two relevant speculative-style decoding baselines for DLMs: SpecDiff[2] and APD[1].
>
> SpecDiff directly applies speculative decoding to the DLM–AR setting, using the DLM to draft in parallel and a stronger ARLM to verify. It therefore serves as the most direct baseline for testing our claim that the large proposal–target gap causes frequent rejection under naive speculative decoding. APD also uses rejection-style decoding, but with a different goal: it prioritizes maximizing DLM throughput while preserving base DLM quality. Accordingly, it uses a lightweight verifier and does not sample correction tokens from the target, keeping outputs biased towards the base DLM.
>
> Due to space constraints, please refer to our response to Reviewer `BVgP` for the detailed experimental settings and full results.
>
> Across the benchmarks, the results are consistent:
> - SpecDiff preserves accuracy relatively well, but its throughput gains are limited because rejection remains frequent, as reflected by both low **Tok./Step** and high **% All Reject**. Since SpecDiff is a direct application of speculative decoding, this provides empirical evidence for our claim that the large gap between $p_D$ and $p_{\mathrm{AR}}$ leads to frequent rejection under the naive speculative decoding setting.
> - APD achieves high throughput, but suffers from substantial quality degradation, consistent with its bias toward the weaker DLM base.
> - PoE-Bridge achieves the strongest overall tradeoff: it remains close to SpecDiff in quality while achieving throughput close to the best APD settings.
>
> These results directly validate our motivation: naive speculative decoding is ineffective in the DLM–AR setting because the proposal–target gap is too large, whereas PoE-Bridge improves efficiency by explicitly bridging that gap.
>
> >**[Q4] Comparison with other acceleration methods for DLMs**
>
> We agree that a broader comparison is valuable. In our view, SpecDiff and APD are the most directly comparable empirical baselines, since they also use rejection-style decoding in the DLM setting. Other acceleration methods target different bottlenecks and are largely complementary to our approach. For example, DLM caching methods [1,3,4] accelerate attention computation through approximate KV caching or more explicit causal structures. These techniques could be applied on top of PoE-Bridge to further improve throughput.
>
> Similarly, TiDAR[5] and SSD[6] mainly accelerate decoding by overlapping drafting and verification, whereas PoE-Bridge improves efficiency by increasing acceptance through explicit proposal–target bridging. We therefore view them as largely orthogonal.
>
> >**Limitations**
>
> We agree that requiring both the AR and DLM to be present during inference increases GPU memory pressure in our current implementation. However, this is a common tradeoff in speculative-style decoding methods with separate proposal and target models, rather than a limitation unique to PoE-Bridge. Also, concurrent single-model approaches such as TiDAR[5] are complementary in this respect and may help alleviate this overhead.
>
> We will clarify this point and add a more detailed limitation discussion in the revised draft. The full discussion is provided in our response to Reviewer `Pma8`. We omit it here due to space constraints.
>
>
> [1] Israel et al. “Accelerating Diffusion LLMs…” NeurIPS, 2025.
>
> [2] Christopher et al. “Speculative Diffusion Decoding…” NAACL, 2025.
>
> [3] Arriola et al. “Block Diffusion…” ICLR, 2025.
>
> [4] Wu et al. “Fast-dLLM…” ICLR, 2026.
>
> [5] Liu et al. “TiDAR…” arXiv, 2025.
>
> [6] Kumar et al. “Speculative Speculative Decoding.” ICLR, 2026.

---

> > ### Author Rebuttal · Reviewer_TLW6 · 2026-04-03
> >
> > Thank you very much for your explanation and apologies for my late acknowledgement as i was tied up with some other work these days. Your responses have helped me understand your work better. I think the method itself is good, but more sophisticated architecture would be more appreciated to help improve the method's applicability in real world deployment. For example, even though speculative decoding uses a draft model, draft models in methods like EAGLE-3 consist of one decoder layer, which is much more lightweight than a AR model. Thus, I'd like to maintain my positive score.

---

### Official Review · Reviewer_BVgP · 2026-03-13

**Soundness:** 3
**Presentation:** 3
**Significance:** 3
**Originality:** 3
**Overall Recommendation:** 4
**Confidence:** 4

**Summary:**

This paper proposes a parallel sampling and speculative decoding framework for diffusion language models, aiming to combine the parallel generation capability of DLMs with the high-quality conditional modeling of AR models. The key motivation is that DLMs can generate tokens in parallel but often produce lower-quality outputs, while AR models capture token dependencies more accurately but suffer from low decoding efficiency. Moreover, the distribution mismatch between DLMs and AR models is typically large, causing naive speculative decoding to frequently reject proposed tokens. To address this issue, the authors introduce a Product-of-Experts (PoE) distribution that lies between the original DLM distribution and the AR distribution. By approximately sampling from this PoE distribution, the method produces candidates that are more compatible with the AR model. The authors further incorporate techniques such as local correction and weighted candidate integration to enable efficient decoding. The proposed method achieves around a 2x speedup while maintaining comparable performance, although a small performance drop is still observed in some cases.

**Compliance With Llm Reviewing Policy:**

Affirmed.

**Final Justification:**

My concerns have been addressed. The authors have shown that, the proposed method achieves better efficiency-accuracy trade-off compared to other baselines.

**Key Questions For Authors:**

Please see the weaknesses. My main concern lies in the peformance drop and baselines.

**Limitations:**

There is NO discussion about the limitations in the paper. It would helpful to include a section about that.

**Strengths And Weaknesses:**

## Strengths

1. The paper presents an interesting observation that directly applying AR-based speculative decoding to DLMs leads to a high rejection rate due to the significant distribution mismatch between the two models. The proposed approach provides a well-motivated solution to bridge this gap between the two decoding paradigms.

2. The authors introduce several practical design choices, including mixed-temperature sampling to mitigate candidate collapse and an elastic rejection window to reduce unnecessary verification computation.

3. The paper conducts systematic explorations of several important hyperparameters, such as the parallel candidate number K
K, the PoE distribution weight w, and the elastic window size V. The experimental results are consistent with the authors’ hypotheses and provide useful insights into the behavior of the method.

## Weaknesses

1. One limitation of the paper is the lack of comparisons with several relevant and important baselines, such as TiDAR [1] and SSD [2]. In addition, the speculative decoding approaches for diffusion language models mentioned in the related work section are not empirically compared. This weakens the support for the performance claims made in the paper.

2. The results in Table 1 show that on more complex benchmarks such as MATH, HumanEval, and MBPP, the method exhibits a noticeable performance drop (around 2–4%). Lossless performance is only observed on relatively simpler tasks such as GSM8K, which may limit the practical applicability of the approach. The authors may compare against more baseline methods, such as other speculative decoding techniques or distillation-based approaches, to better demonstrate that the proposed method achieves a smaller performance degradation.

---

> ### Author Rebuttal · Authors · 2026-03-31
>
> We appreciate the helpful feedback and address the points below.
>
> >**[W1.1] Comparison with concurrent works**
>
> TiDAR[1] and PoE-Bridge both aim to accelerate DLM decoding while keeping outputs close to a stronger AR target, but they do so through different mechanisms. TiDAR is a training-time hybrid model whose speedup comes from overlapping drafting and verification in self-speculative decoding, whereas PoE-Bridge is an inference-time method that improves efficiency by increasing acceptance through a bridge distribution between the proposal and target distributions. SSD[2] is also orthogonal in this sense, since its speedup likewise comes mainly from overlapping drafting and verification.
>
> These directions are also complementary. In principle, PoE-Bridge could combine with either method by incorporating a bridge distribution into the drafting process, potentially further improving throughput via higher acceptance.
>
> We therefore do not view a strict point-to-point empirical comparison as the most informative, especially since the setups differ substantially (e.g., TiDAR uses Qwen3, whereas we use Qwen2.5-based models, and its code is not available). At a high level, however, all of these methods appear to achieve substantial speedup with limited degradation relative to the AR target.
>
> >**[W1.2] Empirical Comparison with speculative decoding approaches for DLM**
>
> We agree that empirical comparison with prior speculative-style DLM decoding baselines would strengthen the paper, and therefore compare PoE-Bridge against two relevant baselines: SpecDiff[3] and APD[4].
>
> SpecDiff directly applies speculative decoding to DLMs, but does not address the large proposal–target gap, leading to frequent rejection and limited speedup (see results). Its multi-step drafting also adds extra DLM compute.
>
> APD instead prioritizes maximizing DLM throughput while preserving base DLM quality. To do so, it uses a lightweight verifier and a rejection rule without sampling a correction token from the target, which keeps its outputs biased toward the base DLM and performance capped by the base DLM quality.
>
> **SpecDiff**: we report its draft-step hyperparameter $T=1,2$, since larger $T$ adds DLM compute without improving accepted tokens in our setup.
>
> **APD**: we report $w=0.5, 0.7$ as representative quality–throughput tradeoffs.
>
> Besides accuracy and throughput, we also report **Tok./Step** (tokens per decoding step) and **% All Reject** (fraction of steps where all drafted tokens are rejected) to quantify rejection behavior.
>
> **MATH**
> ||Acc.|Thrpt.|Tok./Step|% All Reject|
> |---|:-:|:-:|:-:|:-:|
> |Qwen2.5 7B|76.28|46.50|1.00|-|
> |APD (w=0.5)|38.74|93.46|8.76|16.05|
> |APD (w=0.7)|34.02|112.09|11.40|9.94|
> |SpecDiff (T=1)|75.88|66.11|4.55|34.27|
> |SpecDiff (T=2)|73.75|31.90|4.96|26.78|
> |PoE-Bridge (K=1, w=0.3)|73.86|99.49|8.97|10.20|
> |PoE-Bridge (K=4, w=0.3)|74.42|94.94|9.38|8.01|
>
> **HumanEval**
> ||Acc.|Thrpt.|Tok./Step|% All Reject|
> |---|:-:|:-:|:-:|:-:|
> |Qwen2.5 7B|83.54|45.83|1.00|-|
> |APD (w=0.5)|51.83|75.91|6.93|19.13|
> |APD (w=0.7)|48.78|89.61|7.64|14.15|
> |SpecDiff (T=1)|82.72|56.55|4.91|22.74|
> |SpecDiff (T=2)|83.33|31.56|4.70|24.64|
> |PoE-Bridge (K=1, w=0.3)|80.69|84.82|8.30|4.53|
> |PoE-Bridge (K=4, w=0.3)|79.47|76.13|8.56|2.28|
>
> Across the challenging benchmarks MATH/HumanEval—and similarly on MBPP, which we omit here due to space limits—the results show a consistent pattern:
> - APD achieves high throughput, but suffers from substantial quality degradation, consistent with its bias toward the weaker DLM base.
> - SpecDiff preserves accuracy much better, but its throughput gains are limited because rejection remains frequent, as reflected by both Tok./Step and % All Reject.
> - PoE-Bridge achieves the strongest overall tradeoff: staying close to SpecDiff in accuracy while approaching the best APD settings in throughput.
> - Compared with APD, PoE-Bridge remains competitive in throughput even with a larger verifier. We attribute this to our *elastic rejection window*, which reduces the effective cost of AR verification.
>
> >**[W2] Degree of Performance Degradation**
>
> We agree that harder benchmarks are the most meaningful test of practical utility. On MATH and HumanEval, PoE-Bridge degrades relative to the AR target, but much less than APD at comparable speedup, while achieving much higher throughput than SpecDiff with under 3% quality regression.
>
> We are not aware of distillation methods specifically designed for DLM decoding in this setting and would appreciate any relevant references.
>
> >**Limitations**
>
> Due to space limits, please refer to Reviewer `Pma8` for a detailed limitation discussion. In the revised draft, we will include a limitation section, along with the citations and discussions above.
>
> [1] Liu et al. "TiDAR..." arXiv, 2025.
>
> [2] Kumar et al. "Speculative Speculative Decoding." ICLR, 2026.
>
> [3] Christopher et al. "Speculative Diffusion Decoding..." NAACL, 2025.
>
> [4] Israel et al. "Accelerating Diffusion LLMs..." NeurIPS, 2025.

---

> > ### Author Rebuttal · Reviewer_BVgP · 2026-04-02
> >
> > Thanks for the response. My concerns are resolved.

---

### Decision · Program_Chairs · 2026-04-30

**Decision:**

Accept (regular)

**Comment:**

This paper proposes an inference-time framework that bridges the distribution gap between diffusion language models and autoregressive models via a Product-of-Experts intermediate distribution, enabling more efficient parallel decoding with rejection and importance sampling. Practical enhancements include mixed-temperature sampling and elastic rejection windows.
All three reviewers find the core idea well-motivated and technically sound, with a novel formulation for addressing the DLM–AR distribution mismatch. The paper is clearly written and presents consistent empirical results showing speedup over standard DLM decoding while recovering most AR-level quality.
While the contribution is solid and the rebuttal adequate, the limited evaluation scope, non-negligible quality degradation on challenging tasks, and the two-model memory requirement lead to some limitations of the work.